# Pharmacogenetic Analysis Enables Optimization of Pain Therapy: A Case Report of Ineffective Oxycodone Therapy

**DOI:** 10.3390/jpm13050829

**Published:** 2023-05-13

**Authors:** Florine M. Wiss, Céline K. Stäuble, Henriette E. Meyer zu Schwabedissen, Samuel S. Allemann, Markus L. Lampert

**Affiliations:** 1Pharmaceutical Care, Department of Pharmaceutical Sciences, University of Basel, 4056 Basel, Switzerland; celine.staeuble@unibas.ch (C.K.S.); s.allemann@unibas.ch (S.S.A.); markus.lampert@unibas.ch (M.L.L.); 2Institute of Hospital Pharmacy, Solothurner Spitäler, 4600 Olten, Switzerland; 3Biopharmacy, Department of Pharmaceutical Sciences, University of Basel, 4056 Basel, Switzerland; h.meyerzuschwabedissen@unibas.ch

**Keywords:** pharmacogenetics, pain therapy, analgesics, oxycodone, CYP2D6, CYP3A, CYP2C9, therapy failure, adverse drug reaction

## Abstract

Patients suffering from chronic pain may respond differently to analgesic medications. For some, pain relief is insufficient, while others experience side effects. Although pharmacogenetic testing is rarely performed in the context of analgesics, response to opiates, non-opioid analgesics, and antidepressants for the treatment of neuropathic pain can be affected by genetic variants. We describe a female patient who suffered from a complex chronic pain syndrome due to a disc hernia. Due to insufficient response to oxycodone, fentanyl, and morphine in addition to non-steroidal anti-inflammatory drug (NSAID)-induced side effects reported in the past, we performed panel-based pharmacogenotyping and compiled a medication recommendation. The ineffectiveness of opiates could be explained by a combined effect of the decreased activity in cytochrome P450 2D6 (CYP2D6), an increased activity in CYP3A, and an impaired drug response at the µ-opioid receptor. Decreased activity for CYP2C9 led to a slowed metabolism of ibuprofen and thus increased the risk for gastrointestinal side effects. Based on these findings we recommended hydromorphone and paracetamol, of which the metabolism was not affected by genetic variants. Our case report illustrates that an in-depth medication review including pharmacogenetic analysis can be helpful for patients with complex pain syndrome. Our approach highlights how genetic information could be applied to analyze a patient’s history of medication ineffectiveness or poor tolerability and help to find better treatment options.

## 1. Introduction

Chronic back pain is widespread in modern society. It is estimated that more than 19% of individuals over the age of 20 have chronic low-back pain [1]. Common causes of low-back pain are intervertebral disc diseases such as disc hernias [2]. Treatment options for back pain caused by disc hernia include non-pharmaceutical interventions, such as physiotherapy and surgical procedures. Additionally, analgesics are part of the standard procedure pre- and postoperatively [3,4]. Commonly used analgesics are opioids and non-steroidal anti-inflammatory drugs (NSAIDs), but antidepressants, anticonvulsants, and central muscle relaxants are also prescribed [5,6]. The response to drug therapy is often unsatisfactory and varies between individual patients. For some, pain relief is insufficient, while others experience side effects such as nausea, fatigue, dizziness, or respiratory depression [4].

Considering the burden of chronic back pain for the individual patient but also for society, the prescription of effective and safe analgesics is of utmost importance. Possible explanations for interindividual drug response may be differences in age, gender, and environmental factors. Reduced organ function such as renal or hepatic insufficiency may lead to drug accumulation in the circulation and thus promote side effects while drug–drug interactions may affect either efficacy or safety. Furthermore, pharmacogenetic variability may also contribute to differences in drug reactions [7].

Pharmacogenetic considerations in pain management focus on genetic polymorphisms in drug-metabolizing enzymes, drug transporters, and drug targets, and relate these to treatment failure or side effects [7]. For example, genetic variants of cytochrome P450 (CYP) *2D6* influence the bioactivation of codeine and tramadol. Because of the possibility of diminished analgesia, codeine and tramadol should be avoided in patients with a poor metabolizer (PM) status in CYP2D6 [8].

Different expert groups such as the Clinical Pharmacogenetics Implementation Consortium (CPIC) or the Dutch Pharmacogenetics Working Group (DWPG) provide recommendations for opioid treatments under the consideration of genetic polymorphisms [8,9]. However, not only the efficacy of opiates is influenced by pharmacogenetics. Pharmacokinetic and pharmacodynamic properties of non-opioid analgesics or antidepressants for the treatment of neuropathic pain are also affected by genetic variants [10]. It is assumed that pharmacogenetic profiling could contribute to the individualization of analgesic drug regimens, especially in patients with degenerative spinal cord conditions [11].

Despite this knowledge, pharmacogenetic analyses are rarely performed in clinical practice when prescribing analgesics. One possible reason could be limited evidence for the impact of pharmacogenetic analyses on pain reduction in clinical settings [12]. This may be attributed to the fact that in clinical practice, opiates are dosed on the basis of their analgesic effect. It can be assumed that in case of an unfavorable phenotype with increased or reduced opioid metabolism, physicians will adjust and titrate the dosage according to the observed inadequate analgesic effect. This could imply that pharmacogenetic analyses are of little use in pain patients. With this case report, we want to challenge this assumption. We illustrate the benefits of pharmacogenetic analysis in the different classes of analgesics and highlight the complexity of assessing genetic results.

## 2. Materials and Methods

The patient described in this case report was included in an ongoing observational study at our hospital (ClinicalTrials.gov identifier: NCT04154553) approved by the local ethics committee (EKNZ ID: 2019-01452). After obtaining informed consent from the patient, we performed panel pharmacogenotyping, applying the commercial service Stratipharm**^®^** offered by Humatrix AG (Pfungstadt, Germany). This analysis includes more than 30 genes encoding for transport proteins, metabolizing enzymes, and drug targets. Stratipharm**^®^** provides an algorithm-based prediction of the phenotype based on the genotype. In addition to the commercial PGx panel test, we genotyped the rs743966 variant (*UGT2B7* –802 C > T) of the UDP-glucuronosyltransferase (UGT) 2B7 enzyme. The genotyping was performed after DNA extraction from blood samples using the QIAcube and respective chemistry (Qiagen, Hilden, Germany), followed by a restriction fragment length polymorphism (RFLP) assay as previously described [13].

Based on the results of genetic testing and a comprehensive medication review, we compiled a medication recommendation. In addition to gene–drug interactions, we also considered drug–drug interactions and specific drug characteristics. After 1 and 6 months, we conducted a follow-up interview with the patient and asked about changes in medication, as well as the efficacy and tolerability of the drugs given.

## 3. Case Presentation

We report the case of a 34-year-old female patient with chronic pain syndrome. She suffered from back pain due to a disc hernia. The pain radiated to her left leg causing tingling paresthesia in the calf. In addition to the back pain, she suffered from chronic pain in her left ankle. After distortion trauma experienced in the past, the ankle has been operated on 12 times. Due to a significant restriction of daily activities and exhausted conservative therapy options, the patient was referred to our hospital for surgical treatment of the discus hernia. Besides her pain problems, the woman had moderate depression and a history of NSAID-induced non-erosive antrum gastritis.

In a medication reconciliation meeting with a pharmacist, the patient had reported insufficient efficacy of oxycodone, fentanyl, and morphine in the past. For this reason, we performed a pharmacogenetic analysis postoperatively (two days after surgery). At that time, the patient was treated with oxycodone 40 mg daily as baseline medication and additionally with on-demand liquid oxycodone. In parallel, she took the non-opioid analgesics ibuprofen and metamizole (see Table 1 for details of the postoperative medication). Despite extended pain therapy, the patient still complained of severe pain (7–8 of max. 10 points on the numerical rating scale). Because of her depression and the neuropathic component of her pain disorder, she was also treated with venlafaxine, a serotonin and noradrenaline reuptake inhibitor. The woman reported a good response to venlafaxine with respect to her depression, but she did not experience any additional pain relief. Previous therapy with pregabalin was stopped during hospitalization for unknown reasons. Because of the history of NSAID-induced non-erosive antral gastritis and high-dose ibuprofen therapy, she received the proton-pump inhibitor pantoprazole.

The variants in the pharmacogenetic panel test identified the patient as a CYP2C9 intermediate metabolizer (IM, *3 heterozygous), CYP2C8 normal metabolizer (NM, *1), CYP2C19 normal metabolizer (NM, *1), and CYP2D6 intermediate metabolizer (IM, *4 heterozygous). CYP3A5 showed increased activity (IM, *3 heterozygous) (cf. Table 2). Additionally, the patient was identified heterozygote for rs739366 (*UGT2B7*-802CT), rs1799971 (Opioid receptor mu 1; *OPRM1*-118AG), rs4680 (Catechol-O-Methyltransferase; *COMT*-472AG), and rs1045642 (ATP binding cassette subfamily B member 1; *ABCB1*-3435CT) and homozygous for rs2032583 (*ABCB1*-49TT) and rs2032582 (*ABCB1*-2677GG). For these polymorphisms, the algorithm recommends substance-specific phenotype assessment.

## 4. Discussion and Pharmaceutical Assessment

Various authors provided information on the genetic interpretation of different enzymes involved in opioid metabolism in review articles, highlighting that besides genetics physiological factors (gender, weight, age, organ function), environmental factors (diet, tobacco, alcohol) and drug–drug interactions need to be considered for choosing the most suitable substance [7,10,15].

Therefore, we assessed the medication of our patient, taking into account the genetic results, drug interactions, and organ functions. Our patient had normal kidney function and no signs or symptoms of impaired liver function, with all liver parameters within reference ranges. Diminished drug metabolism or elimination due to reduced organ function could thus be excluded. A summary of our pharmaceutical assessments and recommendations for each drug can be found in Table 3.

### 4.1. Assessment of Oxycodone

Despite high doses of oxycodone, the patient suffered from severe postoperative pain. The genetic profile and the analysis of drug–drug interactions provided a hypothesis for the insufficient analgesic effect. Interpretation of the genetic results required a precise understanding of oxycodone metabolism [16]. Figure 1 illustrates the metabolic pathways of oxycodone in the liver.

In detail, oxycodone is metabolized by CYP2D6 and by members of the cytochrome P450 subfamily CYP3A, namely CYP3A4 and CYP3A5. CYP2D6 mediates the O-demethylation of oxycodone to oxymorphone. This pathway accounts for 11% of the total oxycodone degradation. Oxymorphone is characterized by a 40-fold higher affinity for the µ-opioid receptor compared to oxycodone. Furthermore, 45% of oxycodone is N-demethylated by CYP3A4 and CYP3A5 to the inactive metabolite noroxycodone. In turn, oxymorphone and noroxycodone are metabolized by CYP3A4, CYP3A5 and CYP2D6 to noroxymorphone, which is also known for its affinity to the µ-opioid receptor (3-fold higher than oxycodone) [17,18]. It is important to note that the mentioned percentages of oxycodone metabolism represent a population average and may vary among individuals based on their pharmacogenetic profile.

Due to the formation of metabolites with increased affinity for the µ-opioid receptor, one could assume that oxycodone is a molecule that needs to be bioactivated to exert its full analgesic effect. Thus, the Swiss drug label of oxycodone states that a weaker analgesic effect is possible in patients with reduced activity of CYP2D6 [21]. However, due to the weak evidence in the literature, the CPIC refrains from providing recommendations for the drug–gene interaction of oxycodone and CYP2D6 [8].

CPIC’s judgment is based on publications with conflicting results. Clinical studies showed reduced CYP2D6 (PM) activity to result in lower plasma levels of oxymorphone. Nevertheless, the lower oxymorphone plasma levels were not associated with increased oxycodone consumption in CYP2D6 PMs. Furthermore, the authors found no influence of PM phenotype on analgesia or adverse drug reactions [22,23]. Another study, however, showed an increase in cumulative oxycodone consumption 12 h after surgery in CYP2D6 PMs and CYP2D6 IMs compared to CYP2D6 NMs, while pain scores did not differ between the different phenotypes [24]. These results from clinical practice are in contrast to the results from studies in healthy volunteers [25,26]: Two studies showed a lower analgesic effect in CYP2D6 PMs after ingestion of a single dose of oxycodone. In addition, there are some case reports of treatment failure in CYP2D6 PMs [27,28,29].

Possible explanations for these contradictory results regarding CYP2D6 phenotype and oxycodone efficacy are offered by pharmacokinetic studies. Lalovic et al. found that the analgesic effect of oxycodone is mainly mediated by the parent drug oxycodone itself. The two metabolites with increased affinity for the µ-opioid receptor, oxymorphone and noroxymorphone, do not seem to contribute to central analgesia significantly, either because of low concentrations in the circulation (oxymorphone) or lack of transport across the blood–brain barrier (both oxymorphone and noroxymorphone) [18]. Klimas et al. calculated that in CYP2D6 NMs, oxycodone accounts for more than 87% of the analgesic effect, whereas oxymorphone contributes by less than 12%. In CYP2D6 PMs, the involvement of oxycodone increases to more than 97%, and the contribution of oxymorphone decreases to less than 2% [30]. These calculations imply that the quantitative contribution of oxymorphone to the overall analgesic effect is very small, and therefore variations in the CYP2D6 phenotype influence the analgesic effect of oxycodone only to a minor extent.

However, it should be mentioned that most previous pharmacogenetic studies only focused on CYP2D6 when assessing the impact on oxycodone’s analgesic response [22,23]. Even if *CYP3A5* was genotyped, none of the studies examined the effect of the predicted CYP2D6 phenotype in combination with *CYP3A5* variants [24,31], although it is known that variations in *CYP3A5* can influence the metabolism of oxycodone [31]. Briefly, most Europeans are homozygous variants in *CYP3A5* (*3/*3), which results in no enzyme expression in adults. Less than 10% of all Caucasians show a *1 allele with the expression of CYP3A5 [32,33]. A study investigating the influence of the *CYP3A5**1 genotype on oxycodone metabolism showed higher noroxycodone levels and a shift from oxymorphone towards noroxycodone formation compared to patients with a *3/*3 genotype [31]. Considering this effect, we assume that in individuals with the *CYP3A5**1 variant, the influence of the CYP2D6 phenotype on the analgesic effect of oxycodone may become more relevant.

In our patient, we suspected not only reduced oxymorphone levels due to the CYP2D6 IM status but also reduced oxycodone levels due to increased CYP3A5 activity (*1/*3 genotype) and thus a metabolic shift towards inactive noroxycodone. We assume that the increased degradation of oxycodone based on the patient’s genetic predisposition was further amplified by a drug–drug interaction with metamizole. Recent findings showed that metamizole, which was co-administered in our patient, is an inducer of CYP2B6, CYP2C9, CYP2C19, and CYP3A4, and that the induction is mediated by the nuclear receptor constitutive androstane receptor (CAR) [34]. This nuclear receptor is known to also transactivate *CYP3A5* [35]. Importantly, studies with the prototypical pregnane X receptor (PXR) inducer rifampin show that induction is especially high in carriers of *CYP3A5**1 [35]. It is therefore likely that in our case, metamizole induced not only CYP3A4 but also CYP3A5 and therefore promoted the degradation of oxycodone to the inactive metabolite noroxycodone. Our interpretation of the pharmacogenetic results and drug interactions and their effects on the hepatic metabolism of oxycodone are illustrated in Figure 2.

We extended the pharmacogenetic profile of our patient by determination of the *UGT2B7* polymorphism rs7439366. This particular variant has been linked to altered morphine metabolism and increased formation of morphine metabolites. An influence on morphine’s analgesic properties is therefore conceivable [36]. UGT2B7 also plays a role in the metabolism of oxycodone. Indeed, a small proportion of oxycodone is 6-keto reduced and glucuronidated, mediated by UGT2B7 and UGT2B4 [19]. Moreover, UGT2B7 is involved in the glucuronidation and elimination of oxymorphone [20]. Despite this fact and due to the patients’ heterozygosity for this variant, we did not consider the polymorphism to be clinically relevant in this case.

Finally, not only the metabolizing enzymes should be considered when assessing pharmacogenetic results. Target structures are also of relevance. The analgesic effect of oxycodone is primarily mediated by the µ-opioid receptor (encoded by *OPRM1*) [18,37]. The best-studied variant in *OPRM1* is the rs1799971 (A119G) variant, with the G allele associated with reduced mRNA expression and reduced OPRM1 protein levels [38]. A meta-analysis showed that individuals with a AA genotype generally require fewer opioids post-surgery compared to homozygous carriers of GG [39]. Specific to oxycodone, there is evidence that the G allele is associated with lower pain tolerance and the need for higher doses of oxycodone compared to AA [40,41]. In addition, a correlation was found between rs1799971 and gender. Female carriers of a G allele have twice as much pain and a slower recovery rate after lumbar disc herniation as the male carriers [42]. It is therefore possible that the AG genotype in our patient has led not only to a reduced efficacy of oxycodone, but also to an increased perception of pain in general.

In addition, variants in other genes, such as the *ABCB1* gene, encoding for the efflux transporter P-glycoprotein and the *COMT* gene, encoding for the Catechol-O-methyltransferase, may influence the pharmacokinetics and pharmacodynamics of opiates. However, we did not include these genes in our pharmaceutical assessment due to negligible clinical relevance or conflicting evidence in the literature. As an example, neither oxycodone nor oxymorphone is thought to be substrates of P-glycoprotein. Therefore, their transport across the blood–brain barrier is assumed to be independent of genetic variants in the *ABCB1* gene [43,44]. Indeed, the best-studied genetic variants (rs2032582, rs1045642) showed inconsistencies with regard to the analgesic and adverse effects of oxycodone [31,40,45]. Independent of drug metabolism, the rs4680 variant in *COMT* is thought to influence interindividual pain perception. The A allele is associated with a higher pain sensitivity [46].

Taking all mentioned details into account, we assumed that the ineffectiveness of oxycodone could be explained by a combined effect of the decreased activity in CYP2D6, an increased activity in CYP3A, and an impaired drug response at the µ-opioid receptor. Since plasma level determinations for oxycodone are not routinely conducted in clinical practice in our hospital, we did not collect blood serum samples from the patient in this case. This is a limitation of our study as a quantitative analysis of oxycodone and its metabolites in the blood serum could have confirmed our hypothesis of altered oxycodone metabolism.

### 4.2. Assessment of Fentanyl and Morphine

Since the patient also reported treatment failure with other opiates, we decided to include fentanyl and morphine in our analysis. Fentanyl is mainly degraded by CYP3A. Increased CYP3A4 and CYP3A5 activities mediated by the described genetic variants and drug interaction with metamizole were probably responsible for an enhanced inactivation of fentanyl. This assumption is supported by findings in Japanese cancer patients being treated with a fentanyl transdermal reservoir system. Those with a *CYP3A5**1 allele had lower plasma levels of fentanyl and fewer central nervous system (CNS) side effects than those with a *3/*3 genotype [47].

Morphine is metabolized by UGT2B7 and to a lesser extent by UGT1A1 and UGT1A8 to morphine 6-glucuronide and morphine 3-glucuronide. Morphine 6-glucuronide is an active metabolite [48,49]. There are conflicting data linking *UGT2B7* variants to morphine response and efficacy [36,50,51,52]. Considering that our patient is heterozygote in rs7439366, we refrained from further evaluating the *UGT2B7* polymorphism in this case.

Additionally, the analgesic effects of morphine and fentanyl are mediated by the µ-opioid receptor. There is evidence that the G allele found in our patient is associated with increased morphine or fentanyl consumption [53,54,55,56,57,58].

### 4.3. Assessment of Non-Opioid Analgesics

Ibuprofen is a substrate of CYP2C8 and CYP2C9 [59]. Based on her genetic profile, the patient exhibits a normal CYP2C8 activity (*1/*1) and a CYP2C9 IM (*1/*3) status. Reduced activity in CYP2C9 is linked to reduced degradation, prolonged drug half-life, and higher plasma concentrations of ibuprofen. This may result in an increased risk for gastrointestinal side effects [59]. Accordingly, CYP2C9 IM status may have favored the NSAID-induced non-erosive antral gastritis in this case.

In our recommendation, we also took venlafaxine into account. Based on the CYP2D6 status, we suspected a reduced metabolism of its active metabolite, which is linked to an increased risk of therapy failure and side effects [60]. In addition, findings from pharmacogenetic studies imply an association between the homozygous *ABCB1* variant rs2032583 (*ABCB1*-49TT) and antidepressant therapy response due to limited penetration of the blood–brain barrier of antidepressants that are P-glycoprotein substrates [61,62].

### 4.4. Pharmaceutical Recommendation and Outcome

Based on the patient’s genetic profile, we recommended avoiding opiates that are bioactivated by CYP2D6 (codeine and tramadol) or inactivated by CYP3A4/5 (oxycodone and fentanyl). We suggested switching to hydromorphone which is metabolized only to a minor extent by CYP2D6, CYP3A4, or CYP3A5. Hydromorphone is mainly glucuronidated by UGT2B7 in the liver [20,63]. The resulting metabolite, i.e., hydromorphone-3-glucuronide, is considered to be inactive [64]. The variant for *UGT2B7* determined in this case showed no differences in pharmacokinetic parameters of hydromorphone in a study with Taiwanese subjects [65]. Another advantage of hydromorphone may be an additional affinity to the δ-opioid receptor [66]. Hence, the analgesic effect might be less dependent on the rs1799971 (A119G) variant of the µ-opioid receptor.

Besides hydromorphone, tapentadol would also have been an option. Tapentadol is primarily metabolized by glucuronidation involving UGT1A9 and UGT2B7. CYP2C9, CYP2C19, and CYP2D6 are only involved to a minor extent [67]. Additionally, tapentadol is a substance with combined opioid and noradrenergic properties, which make it suitable for the treatment of neuropathic pain [68]. To the best of our knowledge, there are currently no studies that investigated whether genetic variants of *UGT2B7* affect the metabolism and therefore efficacy of tapentadol.

Regarding ibuprofen, the CPIC guidelines recommend using the lowest effective dose for the shortest possible duration for patients with an intermediate metabolizer phenotype of CYP2C9 (activity score = 1). However, due to the patient’s history of ibuprofen-induced gastrointestinal side effects, we recommended avoiding not only ibuprofen but also diclofenac, flurbiprofen, piroxicam, meloxicam, and celecoxib which are all substrates of CYP2C9. We suggested paracetamol in favor of NSAIDs. When switching from oxycodone to hydromorphone, metamizole would also have been an option, as it does not interact with hydromorphone. However, if an NSAID was preferred, naproxen would have been an appropriate choice as it is not metabolized by CYP2C9. We also pointed out that if ibuprofen was still used, the combination with an effective proton-pump inhibitor was necessary. As there was no polymorphism involving *CYP2C19*, pantoprazole at usual doses is a suitable choice.

In contrast to our interpretation of the genetic results, venlafaxine showed good antidepressant efficacy in this case. Therefore, we did not suggest switching to another substance. In addition, we mentioned that pregabalin, which is exclusively renally excreted and therefore not affected by genetic variants in its metabolism, would be a good option for the treatment of neuropathic pain.

According to our recommendations, the treating physicians performed a guidelines-conform opiate rotation and replaced 20 mg of oxycodone 2×/day with 4 mg of hydromorphone 2×/day. Ibuprofen was replaced with paracetamol. The other medications were continued with no changes. In the follow-up interviews after 1 and 6 months, the woman reported adequate pain control with the new analgesic regimen, but again gastrointestinal side effects after a short period of re-intake of ibuprofen.

## 5. Conclusions

Our case report illustrates that an in-depth medication review including pharmacogenetic analysis may be useful for patients with complex pain syndrome. We highlight how genetic information could be applied to analyze a patient’s history of medication ineffectiveness or poor tolerability. In addition, pre-emptive considerations and resulting pharmaceutical recommendations may allow switching to effective and well-tolerated therapy.

As illustrated, the metabolism of opioids is complex, involving multiple enzymes known to be affected by genetics. Consequently, genetic variants in various metabolizing enzymes must be taken into account when evaluating a patient’s pharmacogenetic profile. This procedure requires precise knowledge of the metabolic pathway of each drug. However, not only opioids are influenced by pharmacogenetics. Non-opioid analgesics or antidepressants for the treatment of neuropathic pain may also be affected by genetic variants [10,15]. Further clinical studies are needed to increase the evidence for the relevance of single genetic variants. Furthermore, studies investigating different polymorphisms in combination would be useful, especially for drugs whose metabolism involves several enzymes to a relevant extent.

## Figures and Tables

**Figure 1 jpm-13-00829-f001:**
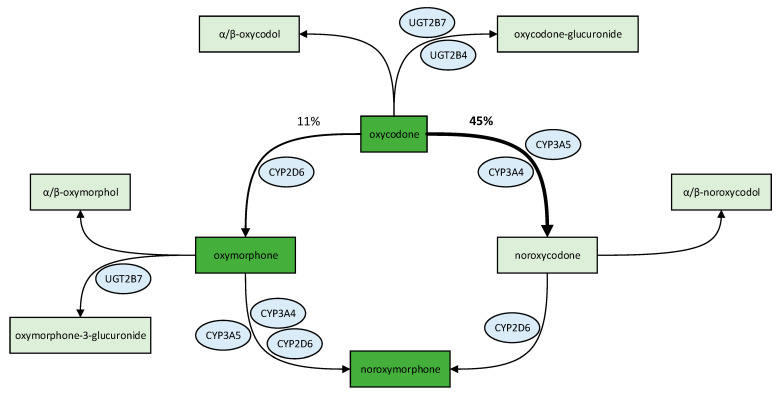
Hepatic metabolism of oxycodone and its metabolites. Substances with strong activity at the µ-opioid receptor are colored dark green. The percentage of oxycodone metabolism (%) represents a population average and has the potential to vary among individuals according to their pharmacogenetic profile [16,17,18,19,20].

**Figure 2 jpm-13-00829-f002:**
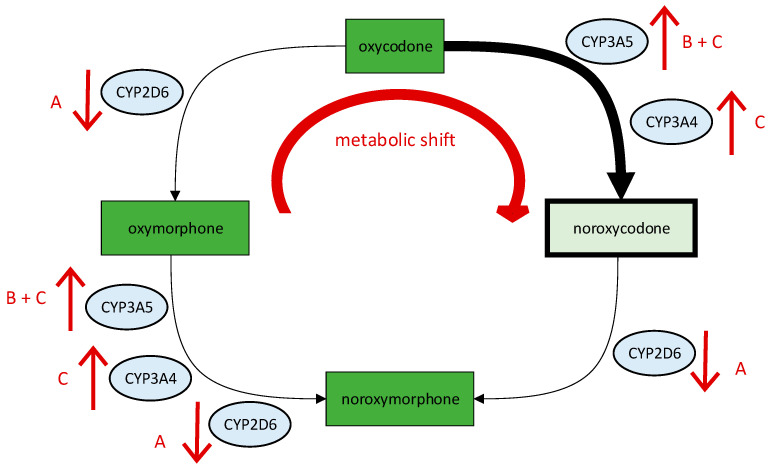
Interpretation of the pharmacogenetic results and drug interactions and their effects on the hepatic metabolism of oxycodone. Substances with strong activity at the µ-opioid receptor are colored dark green; ↑: increased enzyme activity; ↓: reduced enzyme activity; A: reduced activity due to CYP2D6-IM status; B: increased activity due to *CYP3A5* *1/*3 genotype; C: increased activity due to induction by metamizole.

**Table 1 jpm-13-00829-t001:** Postoperative medication.

Substance	Schedule
Oxycodone/Naloxone SR ^a^ 20/10 mg	1-0-1-0
Oxycodone oral Liq ^b^ 10 mg/mL	PRN ^c^ (max. 7 mg/day)
Ibuprofen 600 mg	1-1-1-0
Metamizol gtt ^d^ 0.5 mg/mL	2-2-2-2 mL
Venlafaxine ER ^e^ 150 mg	1-0-0-0
Pantoprazole 40 mg	PRN ^c^ (max. 1 tablet/day)
various laxatives	different

^a^ SR: sustained release, ^b^ Liq: liquid, ^c^ PRN (pro re nata): on demand, ^d^ gtt: drops, ^e^ ER: extended release.

**Table 2 jpm-13-00829-t002:** Selected results of panel pharmacogenotyping and phenotype interpretation [14].

Gene	Variant (Additionally Tested Variants in Gen Locus)	Genotype	Predicted Phenotype (Activity Score)
*CYP2C9*	rs1057910 c.1075A > C (in *3) (rs1799853, rs9332131, rs7900194, rs28371685)	A/C	intermediate metabolizer (AS = 1, reduced function)
*CYP2C8*	(rs10509681, rs11572080, rs1934951)	WT ^a^, *1	n.d. ^d^ (n.d. ^d^)
*CYP2C19*	(rs4244285, rs4986893, rs12248560, rs28399504)	WT ^a^, *1	normal metabolizer (n.d. ^d^)
*CYP2D6*	rs3892097 c.506-1G > A (in *4) rs1065852 c.100C > T (in *4 and *10) (CNV ^c^, rs35742686, rs5030655, rs5030867, rs5030865, rs5030656, rs201377835, rs28371706, rs59421388, rs28371725)	G/A C/T	intermediate metabolizer (AS = 1, reduced function)
*CYP3A5*	rs776746 c.219-237G > A (in *3)	A/G	intermediate metabolizer (n.d. ^d^)
*UGT2B7*	rs7439366 c.802C > T (in *2)	C/T	n.d.^d^ (substance specific)
*OPRM1*	rs1799971 c.118A > G	A/G	n.d. ^d^ (substance specific)
*COMT*	rs4680 c.472G > A (rs165599, rs4646316, rs9332377)	A/G	n.d. ^d^ (substance specific)
*ABCB1*	rs2032583 c.2685 + 49T > C rs1045642 c.3435T > C rs2032582 c.2677G > A or c.2677G > T (rs1128503)	T/T C/T G/G	n.d. ^d^ (substance specific)

^a^ WT: wild type; ^d^ n.d.: not determined, ^c^ CNV: copy number variation.

**Table 3 jpm-13-00829-t003:** Pharmaceutical assessments and recommendations.

Substance	Clinical Effect	Pharmaceutical Assessment	Pharmaceutical Recommendation
Oxycodone	insufficient analgesic efficacy	Metabolic shift to inactive noroxycodone due to decreased activity of CYP2D6, increased activity of CYP3A5 and a drug interaction with metamizole. Impaired drug response at the µ-opioid receptor.	Avoid opiates that are bioactivated by CYP2D6 or inactivated by CYP3A5. Switch to hydromorphone or tapentadol.
Fentanyl	insufficient analgesic efficacy	Increased inactivation due to increased activity of CYP3A5 and a drug interaction with metamizole. Impaired drug response at the µ-opioid receptor.	
Morphine	insufficient analgesic efficacy	Impaired drug response at the µ-opioid receptor.	
Ibuprofen	gastrointestinal side effects	Decreased inactivation due to decreased activity of CYP2C9 and thus an increased risk of gastrointestinal side effects.	Avoid NSAIDs ^a^ that are inactivated by CYP2C9. Switch to paracetamol. Combine NSAIDs ^a^ with a PPI ^b^.
Venlafaxine	good antidepressant but insufficient analgesic efficacy	Reduced metabolism to its active metabolite due to decreased activity of CYP2D6. Limited penetration of the blood–brain barrier due to the *ABCB1* variant.	Continue venlafaxine therapy. Pregabalin as additional therapy option.

^a^ NSAIDs: non-steroidal anti-inflammatory drugs, ^b^ PPI: proton-pump inhibitor.

## Data Availability

The data presented in this study are available on request from the corresponding author. The data are not publicly available for ethical and privacy reasons.

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
