# Peer review of "Pharmacogenetic Analysis Enables Optimization of Pain Therapy: A Case Report of Ineffective Oxycodone Therapy"

_jpm, 2023, doi:10.3390/jpm13050829_

Round 1

Reviewer 1 Report

This is a well written manuscript describing a valuable example of the clinical application of pharmacogenetics to improve the safety and efficacy of treatment for neurological pain. The case is adequately reported, and the investigation and discussion of the authors is excellent and a great example for the scientists in which the genetics and non-genetics factors are involved. I only have minor comments:

- Line 62. Please correct individualization. 

- Line 121 The word identified is repeated in the same sentence. 

- The genes should be italicized in tables and the whole text. 

- Line 173. Please correct CIPIC to CPIC. 

- Line 194. "that" is repeated

- Could the authors mention why CYP3A4 was not genotyped?

- Authors could also mention how the quantification of the drugs and the metabolites could contribute to their findings, or if this was a study limitation?

Reviewer 2 Report

This case report by Florine Wiss, et al. provides a critical investigation of pharmacogenetic variants on oxycodone and analgesics relevant to the case. They organized the case report with a clear introduction to the area of chronic pain and case presentation along with an evidence review of the pharmacogenes that could be potentially relevant for the case, primarily with oxycodone pharmacogenomics. It creates a compelling case report that more research is needed on individual genes and considerations of multiple genes/enzymes involved with the pharmacology of particular medications. The literature analysis focused on oxycodone and all the genes involved was well-organized and included relevant genes/variants of interest. The Figures 1 and 2 nicely helps the reader to understand the metabolism of oxycodone. 

Specific recommendations/comments:

-Genes and variants should be italicized through the paper to avoid confusion between references of the gene versus the enzyme/protein.

-Table 1 “Schedule” column is not clear on how it was prescribed or when the substance was given post-operatively.

-Table 2 would benefit readers if the references to the genotype and predicted phenotypes align with available guidelines such as CPIC or DPWG, specifically with using the star nomenclature for the drug-metabolizing enzymes and activity score scaling for the genes that has defined scales (CYP2C9 and CYP2D6). If the lab reports the data like how it is presented on Table 2, then readers may need additional clarifications how it should be interpreted/applied. This would align with the discussion and literature referenced throughout the text. For example, CYP2C19 genotype *1/*1, normal metabolizer (no activity scoring). CYP2C9 genotype *1/*3, intermediate metabolizer (AS = 1, reduced function).

-Lines 285 to 291 (and table 3) assessment of non-opioid analgesics NSAIDs and the recommendation to avoid based on the CYP2C9 pharmacogenomic results: the CPIC guidelines recommends using the lowest recommended starting dose of ibuprofen for a CYP2C9 intermediate metabolizer phenotype (activity score = 1). I suggest including a similar explanation that was used in lines 170 to 183 on CPIC's recommendations or elaborating further that avoidance is due to other factors beyond genetics such as the patient's history of adverse. If available in the patient's country or to make it applicable to other countries, avoiding meloxicam should also be listed since it is a long-acting NSAID for chronic pain and mentioning a non-CYP2C9 NSAID like naproxen as an alternative. 

Line 301 includes the recommendation to avoid methadone however this recommendation was not elaborated in the paper. Methadone has multiple analgesic properties in addition to the mu-receptor agonism. Although it is widely metabolized by CYP3A4, in vivo studies did not find enzyme variability to be significant compared to the CYP2B6 enzyme. I suggested either removing methadone as the list of CYP3A4 opiates to remove or expand 4.2 Assessment of Fentanyl and Morphine to include methadone.
